# Molecular Biological Methods to Assess Different *Botrytis cinerea* Strains on Grapes

**Louis Backmann** [1,2], **Katharina Schmidtmann** [3], **Pascal Wegmann-Herr** [1], **Andreas Jürgens** [2] **and Maren Scharfenberger-Schmeer** [1,3,*]

1 Institute for Viticulture and Oenology, Dienstleistungszentrum Ländlicher Raum (DLR) Rheinpfalz, Breitenweg 71, D-67435 Neustadt, Germany; louis.backmann@dlr.rlp.de (L.B.); pascal.wegmann-herr@dlr.rlp.de (P.W.-H.)

2 Department of Biology, Chemical Plant Ecology, Technische Universität Darmstadt, Schnittspahnstrasse 4, D-64287 Darmstadt, Germany; juergens@bio.tu-darmstadt.de

3 Hochschule Kaiserslautern, Weincampus Neustadt, Breitenweg 71, D-67435 Neustadt, Germany; katharina.schmidtmann@gmail.com

* Correspondence: maren.scharfenbergerschmeer@hs-kl.de; Tel.: +49-(0)-6321-671-359

**Abstract:** *Botrytis cinerea* is a well-known pathogen that can be challenging to control in crops, such as wine grapes. To adapt to the increasing problems of climate change and strain resistance, it is important to find new methods to detect *Botrytis cinerea* and differentiate strains. These methods include strain differentiation and classification by simple sequence repeats (SSRs) and early detection of the fungus by qPCR. Various strains were analysed using SSR markers and either agarose gel electrophoresis or capillary sequencing via PCR. A sensitive qPCR method was refined to achieve an early detection method for the pathogen. The results demonstrate promising ways to distinguish between strains using both agarose gel electrophoresis and capillary sequencing as well as to detect infection before it becomes visible on grapes. This can be used to further understand and analyse different *Botrytis cinerea* strain characteristics such as laccase activity, regional or annual effects. The early detection method can be used to better prepare growers for an impending infection so that targeted efforts can be made.

**Keywords:** *Botrytis cinerea*; strain differentiation; simple sequence repeat markers; qPCR; early detection

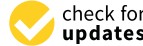



## 1. Introduction

*Botrytis* spp. are a well-known pathogen, infecting over 596 genera of plants, including over 1400 plant species [1]. *Botrytis cinerea* infects 586 genera including tomatoes, strawberries and grapes. Other related species such as *B. allii*, *B. byssoides*, *B. squamosa*, *B. fabae* and *B. gladioli* are pathogens in onions, beans and flowers such as gladioli [2], as well as *B. prunorum*, which causes blossom blight in prunes [3]. *B. cinerea* is a fungus that is found globally, particularly in cold and humid climates [4], as well as in temperate and subtropical climates [5]. It infects grapes (*Vitis vinifera*) and causes significant crop losses such as grey mould or bunch rot, resulting in economic losses of $10–$100 billion per year [6]. *Botrytis* enters grapes through damaged tissues during grape development and remains unnoticed until the grape matures further. This leads to rapid tissue decay in a short period of time, resulting in the harvesting of infected grapes [7]. Additionally, it promotes secondary infection with other pathogens such as *Penicillium expansum* [8]. Infection results in the production of laccase [9,10], a member of the blue copper oxidases [2]. Laccase oxidizes polyphenols into quinones, which then polymerize into brown compounds [11,12]. This process alters the colour of must and wine [13], leading to wine instability and colour degradation. Off-flavours such as geosmin or 1-octen-3-one [14,15] can affect wine quality at concentrations as low as 5% of *Botrytis*-infected grapes [16]. These compounds result

in earthy and mushroom-like flavours. The studies by La Guerche et al. [14,15] have also shown their impact on wine quality. *B. cinerea* growth not only affects the fermentation process in winemaking, but also has a negative impact on wine production [17]. Therefore, it is important to consider prevention measures rather than just treatment. Although there are numerous scientific studies and strategies to minimise the impact of *B. cinerea*, such as applying chemical and biological fungicides on grapes [18] or adding bentonite or oenological tannins to must and wine [13], controlling the disease is challenging due to the various attack pathways and survival strategies of *Botrytis* [6]. These studies have identified a diverse range of potential hosts and both sexual and asexual forms of survival [6,19–21]. Furthermore, ongoing climate change resulting in warmer climates and extreme weather conditions has led to the evolution of more aggressive strains [22]. This has resulted in higher crop losses in wine, even in hot years [23], indicating a problematic future associated with *Botrytis*.

There are two main strategies to combat *B. cinerea*. The first strategy involves applying different treatments to prevent the growth of *Botrytis*, such as botricide, or to reduce the impact of the fungus on the must and wine by using oenological treatments such as active coal, tannins, or flash pasteurisation. To reduce harvest losses and improve the treatment of different *Botrytis* strains, it is important to adapt treatments according to their environmental impact and classify them accordingly. It is crucial to reduce the use of fungicides and other treatments in the wine industry as outlined in the European Green Deal [24] due to the potential risks they pose to human health and the environment [25]. Additionally, *B. cinerea* has demonstrated resistance to several fungicides [26–28]. Microbiological methods, such as a SSR-PCR (simple sequence repeat polymerase chain reaction) or qPCR (real-time polymerase chain reaction), have been shown to be promising tools for gaining a better understanding of the pathogen. For example, Fournier et al. [29,30] developed an SSR-PCR assay to distinguish between different *B. cinerea* strains and to find differences between noble rot and grey mould [30].

The second strategy is to detect *Botrytis* infection at an early stage, before visible signs appear on the grapes. This approach enables more specific and efficient control of the fungus, before the rapid growth of *B. cinerea* causes significant damage. To achieve this, the fungus biomass can be quantified using a highly sensitive method such as qPCR. Quantification of *B. cinerea* by qPCR has already been explored [31]. However, the issue of cross-contamination and the potential impact of different *B. cinerea* strains on qPCR results has not yet been investigated.

This study investigates methods to differentiate between various strains of *B. cinerea*, based on the techniques used by Fournier et al. [29]. Additionally, we evaluate detection methods for quantifying *B. cinerea* in grapes using qPCR, considering the impact of different *B. cinerea* strains and cross-contamination with other pathogen strains present on grapes. Additionally, we want to address the early detection potential of *B. cinerea* using qPCR. The overall objective is to gain a better understanding of the diversity and distribution of *B. cinerea* strains.

## 2. Materials and Methods

### 2.1. Materials

2.1.1. Strain Sampling and Cultivation

During the 2022 harvest season, various *Botrytis cinerea* strains were obtained from different regions and grape varieties (Edenkoben, Göcklingen, Mußbach, Bonn, Mutzig, Barsac, Ihrlingen, Freiburg im Breißgau, Heppenheim, Sternenfels, Oestrich-Winkel, Wollmesheim, Mettenheim, Zeltingen-Rachtig, Table 1). The strains were collected from sporulating berries. Additionally, strains from the DLR RLP Phytomedicine collection from previous harvest seasons (Grünstadt) and one DSMZ Strain 877 (N51, DSMZ German Collection of Microorganisms and Cell Cultures GmbH, Braunschweig, Germany) were used. The strains were cultivated on 2% Biomalz Agar (BA) plates at 25 °C. Single spore isolates were created and refreshed every 2–4 weeks. *Yarrowia lipolytica* yeast was cultured on potato dextrose

agar (PDA) plates and incubated at 30 °C for 2 days. *Cladosporium* sp., *Trichothecium roseum* and *Penicillium expansum* were grown on BA.

**Table 1.** Collection sites of the tested strains. The accuracy of the location is about 10–20 m.

| Strain | Date of Collection | Location (°N, °E) | Region |
|---|---|---|---|
| 1 | 7 October 2022 | 49.285899, 8.194810 | Edenkoben |
| 2 | 7 October 2022 | 49.285899, 8.194810 | Edenkoben |
| 3 | - | DSMZ Strain | - |
| 4 | 11 October 2013 | 49.553507, 8.154937 | Grünstadt |
| 5 | 7 September 2022 | 49.153390, 8.022599 | Göcklingen |
| 6 | 7 September 2022 | 49.153390, 8.022599 | Göcklingen |
| 7 | 25 September 2022 | 49.368001, 8.186013 | Mußbach |
| 8 | 25 September 2022 | 49.368001, 8.186013 | Mußbach |
| 9 | 6 October 2022 | 50.729375, 7.067771 | Bonn |
| 10 | 25 October 2022 | 49.293299, 8.208207 | Edenkoben |
| 11 | 25 October 2022 | 49.293299, 8.208207 | Edenkoben |
| 12 | 25 October 2022 | 49.293299, 8.208207 | Edenkoben |
| 13 | 25 October 2022 | 49.285899, 8.194810 | Edenkoben |
| 14 | 25 October 2022 | 49.285899, 8.194810 | Edenkoben |
| 15 | 25 October 2022 | 49.285899, 8.194810 | Edenkoben |
| 16 | 28 October 2022 | 49.222630, 8.112190 | Mußbach |
| 17 | 28 October 2022 | 49.222630, 8.112190 | Mußbach |
| 18 | 28 October 2022 | 49.222630, 8.112190 | Mußbach |
| 19 | 4 October 2022 | 48.539542, 7.472217 | Mutzig |
| 20 | 4 October 2022 | 48.539542, 7.472217 | Mutzig |
| 21 | 4 October 2022 | 48.539542, 7.472217 | Mutzig |
| 22 | 25 September 2022 | 49.368001, 8.186013 | Mußbach |
| 23 | 28 October 2022 | 44.612330, −0.321527 | Barsac |
| 24 | 7 September 2022 | 48.054682, 7.624740 | Ihrlingen |
| 25 | 7 September 2022 | 48.054682, 7.624740 | Ihrlginen |
| 26 | 7 September 2022 | 48.054682, 7.624740 | Ihrlingen |
| 27 | 7 September 2022 | 47.979199, 7.833400 | Freiburg |
| 28 | 7 September 2022 | 47.979199, 7.833400 | Freiburg |
| 29 | 7 September 2022 | 47.979199, 7.833400 | Freiburg |
| 30 | 19 September 2022 | 49.625441, 8.647692 | Heppenheim |
| 31 | 19 September 2022 | 49.625797, 8.648443 | Heppenheim |
| 32 | 23 September 2022 | 49.047806, 8.842546 | Heppenheim |
| 33 | 23 September 2022 | 49.047806, 8.842546 | Sternenfels |
| 34 | 23 September 2022 | 49.047806, 8.842546 | Sternenfels |
| 35 | 23 September 2022 | 49.043591, 8.850927 | Sternenfels |
| 36 | 23 September 2022 | 49.043591, 8.850927 | Sternenfels |
| 37 | 15 September 2022 | 50.006921, 8.001069 | Oestrich-Winkel |
| 38 | 15 September 2022 | 50.006921, 8.001069 | Oestrich-Winkel |
| 39 | 15 September 2022 | 50.006921, 8.001069 | Oestrich-Winkel |
| 40 | 15 September 2022 | 50.006921, 8.001069 | Oestrich-Winkel |
| 41 | 15 September 2022 | 50.006921, 8.001069 | Oestrich-Winkel |
| 42 | 7 September 2022 | 49.183386, 8.089927 | Wollmesheim |
| 43 | 7 September 2022 | 49.183386, 8.089927 | Wollmesheim |
| 44 | 7 September 2022 | 49.183386, 8.089927 | Wollmesheim |
| 45 | 25 September 2022 | 49.368001, 8.186013 | Mußbach |
| 46 | 25 September 2022 | 49.368001, 8.186013 | Mußbach |
| 47 | 15 September 2022 | 49.735226, 8.327686 | Mettenheim |
| 48 | 26 September 2022 | 49.952237, 7.026950 | Zeltingen-Rachtig |
| 49 | 26 September 2022 | 49.952237, 7.026950 | Zeltingen-Rachtig |
| 50 | 30 October 2022 | 49.225930, 8.104260 | Mußbach |
| 51 | 30 October 2022 | 49.225930, 8.104260 | Mußbach |
| 52 | 30 October 2022 | 49.225930, 8.104260 | Mußbach |
| 53 | 30 October 2022 | 49.225930, 8.104260 | Mußbach |
| 54 | 15 September 2022 | 50.014349, 8.002587 | Oestrich-Winkel |
| 55 | 15 September 2022 | 50.014349, 8.002587 | Mörzheim |
| 56 | 7 September 2022 | 49.155096, 8.071153 | Mörzheim |
| 57 | 7 September 2022 | 49.155096, 8.071153 | Mörzheim |
| 58 | 7 September 2022 | 49.155096, 8.071153 | Mörzheim |
| 59 | 26 September 2022 | 49.913841, 7.049492 | Zeltingen-Rachtig |

### 2.1.2. Preparation of Field Samples

Grape samples of the Pinot Noir and Riesling varieties were obtained at different times during the harvest season. A total of 50 grapes were collected throughout the vineyard.

An average sample of the vineyard was obtained by counting and packing three sets of 100 berries into separate bags, which were then stored in the freezer. The berries were crushed, using an Ultra-thorax (MICCRA, Buggingen, Germany) to homogenize the sample prior to use. For DNA extraction, 0.5 g of each sample was centrifuged at $13,000\times g$ for 15 min and the resulting pellet was retained. Subsequently, $8 \times 10^6$ cells of *Yarrowia lipolytica* were added to the sample, as an internal positive control for qPCR, and the mixture was centrifuged again for 15 min at $13,000\times g$. The resulting pellet was retained and subject to extraction following the RED Extract Plant PCR-Kit (Merck KGaA, Darmstadt, Germany) protocol.

### 2.2. Methods

#### 2.2.1. DNA Extraction

The strains and samples were extracted using the RED Extract Plant PCR kit from Merck KGaA, Darmstadt, Germany. Strains were harvested from agar plates and samples were obtained from grape berries. Pre-tests for this research have shown that this kit is an effective tool for quickly obtaining fungal DNA. Specifically, 100 μL of extraction solution was added to the prepared sample, mixed by vortexing, and incubated in a heat block at 95 °C for 10 min. Subsequently, 100 μL of dilution solution was added mixed by vortexing and the resulting mixture was used for further analysis. After centrifugation at $13,000\times g$ for 10 min, the soluble phase was retained and stored in the freezer for future use.

#### 2.2.2. qPCR—Preparation of Cultivated Botrytis Samples—Cross-Contamination

*Botrytis cinerea* strains were grown to promote sporulation and then removed from the plates. *Penicillium expansum*, *Trichothecium roseum* and *Cladosporium* sp. were also grown to promote sporulation and added separately to the *B. cinerea* probes. DNA was extracted using the method described above.

#### 2.2.3. qPCR—Preparation of Standard Curves

During the experiment, seven strains of *B. cinerea* were used to create the standard curve (Table 2). To create the qPCR standard curve for *B. cinerea*, a solution of $10^7$ spores/mL was obtained by filtering *B. cinerea* mycel from agar plates after sporulation through a sieve and adding water. The spore count was determined using a light microscope and Neubauer counting chamber, and a solution of $10^7$ spores/mL was created. Serial dilutions were used to obtain spore solutions ranging from $10^2$ to $10^7$ spores/mL. DNA was extracted using the RED Extract protocol. The *Y. lipolytica* standard curve was obtained in a similar manner to the *B. cinerea* standard curve, with the addition of an additional $10^8$ spores/mL solution.

**Table 2.** *Botrytis cinerea* strains used for qPCR and their location and date of collection.

| Strain | Location | Date of Collection |
|---|---|---|
| Strain 1 | Rupperstberg, Germany | 9 September 2008 |
| Strain 2 | Göcklingen, Germany | 17 September 2021 |
| Strain 3 | Italy | 2 February 2011 |
| Strain 4 | Wachenheim, Germany | 16 September 2021 |
| Strain 5 | Geinsheim, Germany | 31 August 2021 |
| Strain 6 | Laumersheim, Germany | 30 August 2021 |
| Strain 7 | Deidesheim, Germany | 22 September 2009 |

#### 2.2.4. qPCR—Run

The adapted version of the primer (Table 3) was used to measure all probes and standard curves in triplets, following the protocol described in Diguta et al. [31]. The qPCR protocol consisted of a hold stage at 95 °C for 3 min, followed by a two-step PCR (95 °C 15 s, 65 °C 30 s—40 cycles), melting curve (90 °C 15 s, 50 °C 60 s, 95 °C 1 s). The resulting ct-values of the probes were analysed using the qPCR standard curves.

**Table 3.** *Botrytis cinerea* Targeting Seq. Ribosomal Region 28S, 18S (Suarez et al. [32]). The original primer pair (QBc) was extended (QBc10).

| Primer | Sequence |
|--------|----------|
| QBc | F: GCTGTAATTTCAATGTGCAGAATCC<br>R: GGAGCAACAATTAATCGCATTTC |
| QBc10 | F: GCTGTAATTTCAATGTGCAGAATCCTGTCCCCGGT<br>R: GGAGCAACAATTAATCGCATTTCAAACATGCTG |

### 2.2.5. qPCR—Early Detection Method—Limit of Detection

In the experiment, 300 berries per variant (Thompson Seedless, Peru) were used. A control with 0 spores/berry was compared to a variant with 10,000 spores/berry. The berries were "surface-sterilized" for 30 s in a 70% ethanol washing solution. The remaining ethanol was washed out using sterilized water. Subsequently, a 10 µL droplet with 10,000 spores and a 10 µL droplet containing water (control) were applied to each berry. Every 24 h, $3 \times 10$ berries were collected per variant. The berries were crushed using an Ultra-Thorax (MICCRA, Buggingen, Germany) and 0.5 g were used for DNA extraction. qPCR was performed as previously described. The experiment was continued until first sporulation was visible.

### 2.2.6. SSR-PCR—PCR Run

A total of eight different simple sequence repeat markers, established by Fournier et al. [29], were used to perform PCR. The primers were tested for their size range at different temperatures and in different pairs to obtain optimal sets for a multiplex set of primers. The resulting multiplex sets were 1, 2, 4 and 3, 5, 6 at 50 °C and 7, 10 at 60 °C (Table 4). PCR was performed following a standard procedure: The reaction tubes were prepared by adding 10 µL of RED Extract (Sigma-Aldrich, St. Louis, MO, USA), 4 µL MilliQ water, 1 µL Forward/Reverse Primer each, and 4 µL sample DNA. The PCR program consisted of 36 cycles of denaturation at 94 °C for 30 s, annealing at a primer dependent temperature (50 °C, 60 °C) for 30 s, and elongation at 72 °C for 30 s for a total of 36 cycles, preceded by initial denaturation at 94 °C for 3 min and followed by a final elongation at 72 °C for 10 min. The PCR was performed using an Eppendorf MasterCycler personal 5332 (Eppendorf SE, Hamburg, Germany). The PCR samples were stored at −20 °C in the freezer or at 4 °C in the fridge for later use.

**Table 4.** Primer sequences used for SSR-PCR followed by agarose gel electrophoresis.

| Primer | Sequence | Temperature and Primer Mix | Expected Size Range (Fournier et al. [29]) |
|--------|----------|---------------------------|-------------------------------------------|
| Bc1 | F: AGGGAGGGTATGAGTGTGTA<br>R: TTGAGGAGGTGGAAGTTGTA | Primer Mix 1 (50 °C) | 245–281 |
| Bc2 | F: CATACACGTATTTCTTCCAA<br>R: TTTACGAGTGTTTTTGTTAG | Primer Mix 1 (50 °C) | 161–205 |
| Bc3 | F: GGATGAATCAGTTGTTTGTG<br>R: CACCTAGGTATTTCCTGGTA | Primer Mix 2 (50 °C) | 197–229 |
| Bc4 | F: CATCTTCTGGGAACGCACAT<br>R: ATCCACCCCCAAACGATTGT | Primer Mix 1 (50 °C) | 98–125 |
| Bc5 | F: CGTTTTCCAGCATTTCAAGT<br>R: CATCTCATATTCGTTCCTCA | Primer Mix 2 (50 °C) | 143–163 |
| Bc6 | F: ACTAGATTCGAGATTCAGTT<br>R: AAGGTGGTATGAGCGGTTTA | Primer Mix 2 (50 °C) | 88–158 |
| Bc7 | F: CCAGTTTCGAGGAGGTCCAC<br>R: GCCTTAGCGGATGTGAGGTA | Primer Mix 3 (60 °C) | 113–131 |
| Bc10 | F: TCCTCTTCCCTCCCATCAAC<br>R: GGATCTGCGTGGTTATGA | Primer Mix 3 (60 °C) | 158–189 |

### 2.2.7. SSR-PCR—Agarose Gel Electrophoresis

To perform agarose gel electrophoresis, 3.1 g agarose (AppliChem GmbH, Darmstadt, Germany) were added to 100 mL of 10× TAE buffer (AppliChem GmbH, Darmstadt, Germany) and microwaved at 600 W for 3 to 4 min until the mixture became smooth. After

cooling the mixture to −50 °C, 10 μL GelRed (GeneON, Ludwigshafen, Germany) was added. The gel was prepared by adding 5 μL of the PCR probes and 2 μL of the ladder (Thermo Fisher Scientific, Waltham, MA, USA) and running the reaction at 90 V for 2 h and 45 min. The resulting gel was captured using a UV camera lens.

### 2.2.8. SSR-PCR—Evaluation of the Gel

To assess the resulting gel bands, we measured the bands on the captured images using an ImageJ software tool (version 1.51) [33] and compared them with the ladder. We compared resulting PCR product sizes to obtain a set of band sizes for each tested strain.

### 2.2.9. SSR-PCR—Capillary Sequencer

A total of eight simple sequence repeat markers, established by Fournier et al. [29], were used to perform PCR. The primers were tested at different temperatures and in different pairs to obtain optimal sets for a multiplex set of primers. The obtained multiplex sets were Bc1, Bc5, Bc10; Bc2, Bc3, Bc6 and Bc4, Bc7, Bc9 at 60 °C. The procedure followed the protocol by Huber et al. [34]. A KAPA2G Fast Multiplex PCR Kit (KAPABIOSYSTENS, Wilmington, MA, USA) was used to conduct the multiplex PCR, which included up to 10 primer pairs with fluorescent labels (forward primer coupled with HEX, ROX, TAMRA or FAM (Table 5). The PCR program consisted of 30 cycles of denaturation, annealing and elongation, with initial denaturation at 95 °C for 3 min, denaturation at 95 °C for 15 s, primer annealing at 60 °C for 30 s, elongation at 72 °C for 30 or 50 s, and final elongation at 72 °C for 3 min. The length analyses of the fragments were performed using a 3130xl Genetic Analyzer (Applied Biosystems, Darmstadt, Germany) and the corresponding GeneMapper 4.0. software. The polymer used was POP7™ (Thermo Fisher Scientific, Waltham, MA, USA). The PCR samples were stored at 4 °C in the fridge or in the freezer (−20 °C) for later use.

**Table 5.** Table of the primers by Fournier et al. [29] and the used fluorescence labels. The forward primers of each primer pair were coupled with a fluorescent dye (HEX, ROX, TAMRA or FAM) to allow multiplex PCR.

| Primer | Sequence | Label | Primer Mix |
|--------|----------|-------|------------|
| Bc1 | F: AGGGAGGGTATGAGTGTGTA | Rox | Primer Mix 1 |
| | R: TTGAGGAGGTGGAAGTTGTA | - | |
| Bc2 | F: CATACACGTATTTCTTCCAA | Tamra | Primer Mix 2 |
| | R: TTTACGAGTGTTTTTGTTAG | - | |
| Bc3 | F: GGATGAATCAGTTGTTTGTG | 6-Fam | Primer Mix 2 |
| | R: CACCTAGGTATTTCCTGGTA | - | |
| Bc4 | F: CATCTTCTGGGAACGCACAT | Rox | Primer Mix 3 |
| | R: ATCCACCCCCAAACGATTGT | - | |
| Bc5 | F: CGTTTTCCAGCATTTCAAGT | 6-Fam | Primer Mix 1 |
| | R: CATCTCATATTCGTTCCTCA | - | |
| Bc6 | F: ACTAGATTCGAGATTCAGTT | 6-Fam | Primer Mix 2 |
| | R: AAGGTGGTATGAGCGGTTTA | - | |
| Bc7 | F: CCAGTTTCGAGGAGGTCCAC | Hex | Primer Mix 3 |
| | R: GCCTTAGCGGATGTGAGGTA | - | |
| Bc9 | F: CTCGTCATAACCACGCAGAT | 6-Fam | Primer Mix 3 |
| | R: GCAAGGTCTCGATGTCGATC | - | |
| Bc10 | F: TCCTCTTCCCTCCCATCAAC | Hex | Primer Mix 1 |
| | R: GGATCTGCGTGGTTATGA | - | |

### 2.2.10. Statistical Analysis

Statistical analysis was performed using XLSTAT 2016. Principal component analysis was conducted, assuming the data can be described on a lower dimension. Paired t-tests were conducted after checking for normal distribution of the data set.

## 3. Results

### 3.1. qPCR Cross-Contamination

During an infection with *Botrytis cinerea*, cross-contaminations with different fungi can occur, which may affect the quantification of *Botrytis* biomass. To ensure primer specificity, the primers used in qPCR were tested against fungi that are commonly found on grapes including *Penicillium expansum*, *Trichothecium roseum* and *Cladosporium* sp. *B. cinerea* and *P. expansum* were extracted separately and together and then tested for amplification using PCR and qPCR. The pure *Penicillium* probe did not show any bands in the PCR test, but qPCR showed a slight amplification of *Penicillium* using the original primer set. Therefore, the primer set was adapted by making the forward and reverse primers 10 bp longer on the 3′-end, according to the known genome sequence in Ensembl (version 109). Further testing revealed no amplification *Penicillium* or other tested fungi. Additionally, the primer exhibited no cross-reaction with the tested fungi (Figure 1). Both primer sets were free of primer dimers or by-products in the melting curve.

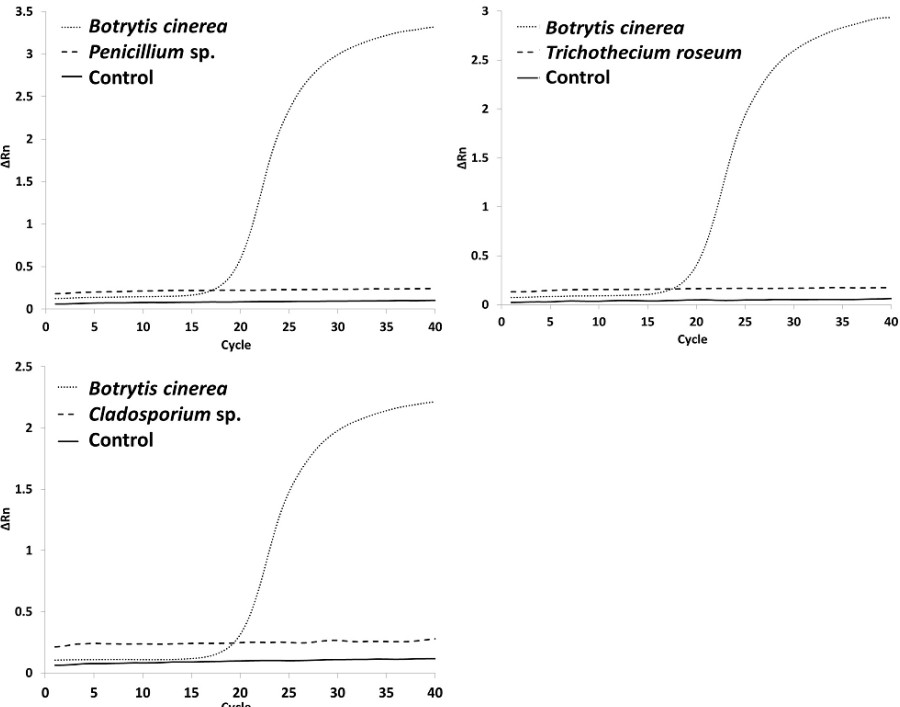

**Figure 1.** Cross-contamination results of the QBc10 Primer. In every run, the primer was tested against a *Botrytis cinerea* strain, either *Penicillium expansum*, *Trichothecium roseum* or *Cladosporium* sp. and a negative control.

### 3.2. Effect of Different Botrytis Strains on the Standard Calibration Curve

To investigate the potential impact of different *B. cinerea* strains on qPCR efficiency, we prepared and counted spore suspensions under a light microscope to obtain both high and low concentrations of spores per millilitre. The strains used were: strain 1 (Ruppertsberg, Germany, 2008), strain 2 (Göcklingen, Germany, 2021), strain 3 (Italy, 2011), strain 4 (Wachenheim, Germany, 2021), strain 5 (Geinsheim, Germany, 2021), strain 6 (Laumersheim, Germany, 2021). The spore suspensions were extracted and quantified against a standard curve using the same *B. cinerea* strains for every measurement, specifically strain 7 from Deidesheim, Germany, 2009. The primer efficiency ranged from 94% to 96%. The results were then compared with the counted cell number as shown in Figure 2. Most of the *Botrytis* strains used for the standard calibration curve were within a reasonable variance range compared to the manual counting, considering the limitations of obtaining exact results using the counting chamber. However, strain 3 showed a greater variance of up to 23.5%.

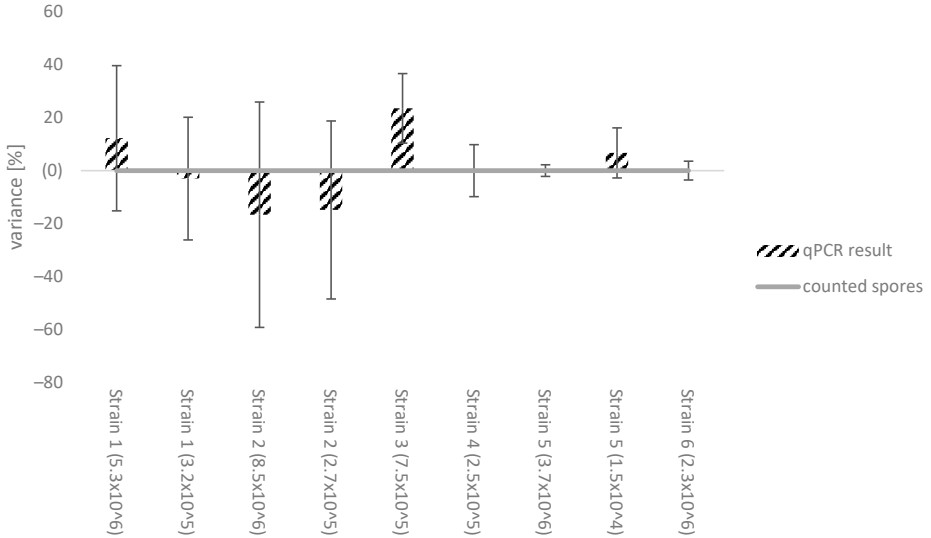

**Figure 2.** Comparison of the counted spore number to the different qPCR results using the same *Botrytis cinerea* strain as standard calibration curve (strain 7, Deidesheim, Germany, 2021). Strains used: strain 1 (Ruppertsberg, Germany, 2008), strain 2 (Göcklingen, Germany, 2021), strain 3 (Italy, 2011), strain 4 (Wachenheim, Germany, 2021), strain 5 (Geinsheim, Germany, 2021), strain 6 (Laumersheim, Germany, 2021) The results were normalized towards the counted spore number (Neubauer counting chamber). In some cases, two different spore concentrations representing a high and low spore number were analysed.

### 3.3. qPCR as an Early Detection Method—Limit of Detection

In the experiment, early signs of infection were observed on days 3 and 4, although no sporulation was visible on the berry. The detected biomass increased during the infection period and sporulation became visible after seven days (see Figure 3). Paired t-tests were performed for every individual between the spore and control variant (*** = $p < 0.0001$). The detection limit of the qPCR was 100 spores.

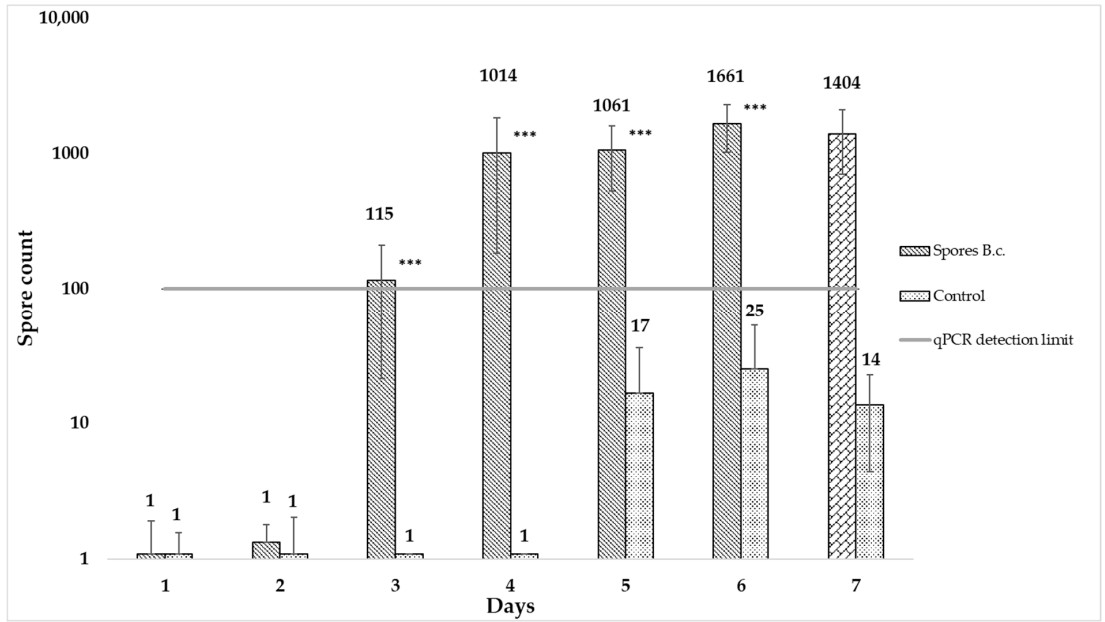

**Figure 3.** Early detection of *Botrytis cinerea* (*B.c.*) on table grapes. Berries were inoculated with 10,000 spores/berry and compared to a control. Every 24 h, 3 × 10 berries were collected per variant, crushed, 0.5 g were extracted and qPCR was performed. The experiment continued until sporulation was visible on the grapes (paired *t*-test; *** = $p < 0.0001$).

### 3.4. PCR Method Validation—Size Range of Primer Sets and Composition

The amplified bands of the different simple sequence repeat markers were of varying sizes. A total of 59 strains underwent testing through capillary sequencer and agarose gel analysis and the results were compared (Table 6). For the agarose gel electrophoresis, the resulting band sizes of the strains were compared to each other using the results of the imageJ comparison (See Figure 4 as an example). Every strain with a different composition of band sizes than a previous one was assigned a new identity. The resolution of the imageJ comparison was around 10 bp. The primer pair Bc9 was not used for agarose gel electrophoresis since it provided no further information and inhibited the primer mixes. For the capillary sequencer, the resulting peaks in the histogram at their respective base pair size were compared (see Figure 5 as an example). Every strain with a different composition of primer pair sizes was assigned with a new identity. Both methods were able to distinguish a large proportion of the tested strains.

**Table 6.** Comparison of different methods to distinguish *Botrytis cinerea* strains using simple sequence repeat markers. In total, 59 strains were analysed by PCR followed by either an agarose gel electrophoresis or capillary sequencer. For every primer pair and method, the resulting basepair ranges of the strains are shown.

| Primer | Capillary Sequencer | Agarose Gel | Fournier et al. [29] |
|---|---|---|---|
| Bc1 | 219–265 | 195–256 | 245–281 |
| Bc2 | 144–183 | 132–180 | 161–205 |
| Bc3 | 213–221 | 189–240 | 197–229 |
| Bc4 | 118–127 | 107–132 | 98–125 |
| Bc5 | 117–162 | 135–169 | 143–163 |
| Bc6 | 109–125 | 105–130 | 88–158 |
| Bc7 | 111–119 | 105–120 | 113–131 |
| Bc9 | 146–147 | / | 150–194 |
| Bc10 | 178–189 | 165–200 | 158–189 |

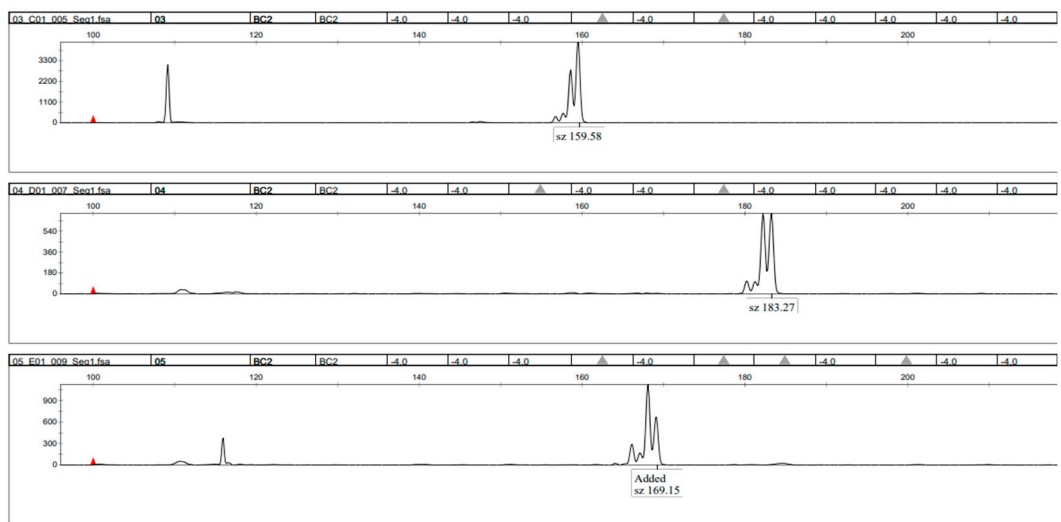

**Figure 4.** Sample picture of the capillary sequencer result. Labelled peaks in the diagram represent primer pair products in their respective basepair size. The tested strains were manually analysed after the experiment to ensure correct measurements of the software.

The tested primer pairs vary on how they affect the differentiation of the strains. Principal component analysis (PCA) was conducted and shows that some of the primers have a greater variety regarding the band sizes detected by SSR-PCR (Figure 6). For example, Bc1 and Bc5 contribute greatly to strain differentiation while Bc6 and Bc9 do not contribute much to strain differentiation.

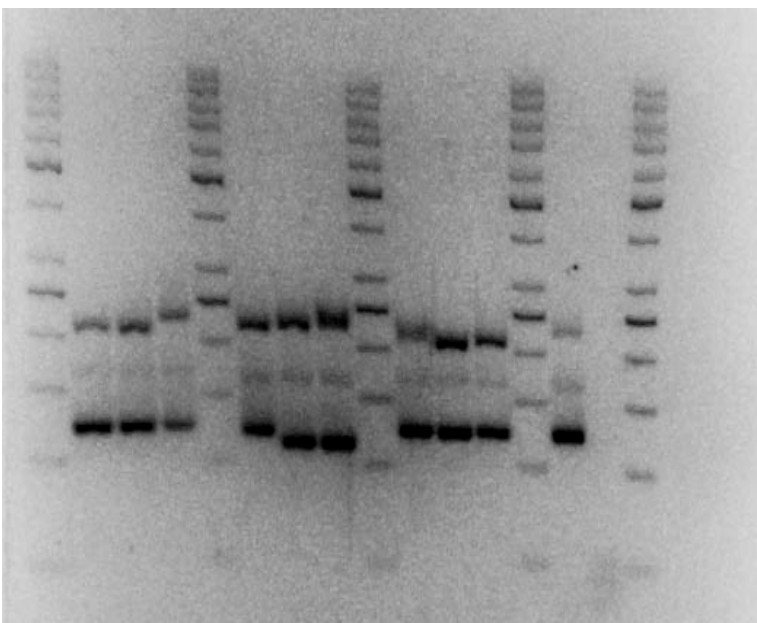

**Figure 5.** Sample picture of the Agarose gel electrophoresis. 10 *Botrytis cinerea* strains, one primer mix containing the primer pairs Bc1, Bc2 and Bc4. A normal run consisted of 10 strains and 1 negative control tested against 1 primer mix (either 1–2–4, 3–5–6 or 7–10). Runtime 2 h 45 m at 90 V. (picture was brightened for better clarity, original picture in Supplementary Marerials (Figure S15); contrast +1.51, Brightness +100, Light +100, Edited with Adobe Lightroom Classic CC).

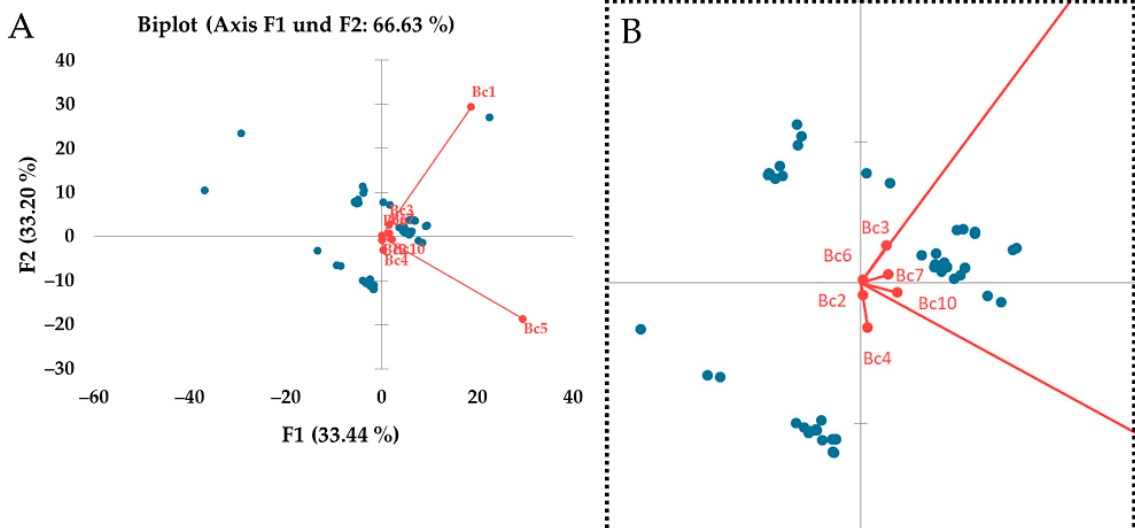

**Figure 6.** Principal component analysis (PCA) of the tested strains in SSR-PCR. The influence of the used primer pairs on the differentiation of the strains can be seen (**A**). Zoomed in perspective of the PCA (**B**).

### 3.5. Regional Differences between Strains

The strains were collected throughout different regions. Most of the regions showed a high diversity of strains regarding the diverging fragment sizes of the SSRs. Differences within one region could be observed in the following regions, given the number of strains: Edenkoben ($n = 8$), Mußbach ($n = 12$), Ihrlingen ($n = 3$), Freiburg ($n = 3$), Heppenheim ($n = 2$), Sternenfels ($n = 5$), Oestrich-Winkel ($n = 6$), Wollmesheim ($n = 3$), Mörzheim ($n = 4$), Mutzig ($n = 3$), Zeltingen-Rachtig ($n = 3$). There was one region where the tested strains were similar to each other: Göcklingen ($n = 2$).

Within their respective regions, some, but not all of the strains were similar to each other (two or fewer differences detected by capillary sequencer: Edenkoben, Mußbach, Freiburg, Oestrich-Winkel, Mörzheim, Mutzig and Göcklingen. In contrast, all of the strains from the other regions differ from one another, within their respective regions (more than two differences detected by capillary sequencer): Ihrlingen, Heppenheim, Wollmesheim, Zeltingen-Rachtig. More regions were tested, but the number of isolated strains was $n = 1$: Bonn, Barsac, Grünstadt, Mettenheim and the DSMZ strain.

The regions were compared to each other by the diverging fragment sizes of each strain. Bonn showed the greatest difference from every other tested strain ($n = 1$, 85.8%), whereas Göcklingen showed the smallest difference ($n = 2$, 45.6% $\pm$ 0%; Table 7).

**Table 7.** Differences of the tested regions by SSRs, compared to the tested strain population ($n = 59$).

| Location | Difference from All Strains |
|---|---|
| Bonn ($n = 1$) | 85.8% |
| (DSMZ) ($n = 1$) | 73.5% |
| Ihrlingen ($n = 3$) | 66.5% $\pm$ 6.2% |
| Zeltingen-Rachtig ($n = 3$) | 62.4% $\pm$ 9.0% |
| Mörzheim ($n = 4$) | 62.4% $\pm$ 12.1% |
| Freiburg ($n = 3$) | 58.9% $\pm$ 4.1% |
| Grünstadt ($n = 1$) | 58.3% |
| Edenkoben ($n = 8$) | 57.0% $\pm$ 8.4% |
| Barsac ($n = 1$) | 55.3% |
| Wollmesheim ($n = 3$) | 53.5% $\pm$ 2.3% |
| Mußbach ($n = 12$) | 53.2% $\pm$ 5.6% |
| Oestrich-Winkel ($n = 6$) | 52.3% $\pm$ 5.2% |
| Sternenfels ($n = 5$) | 52.0% $\pm$ 8.8% |
| Mutzig ($n = 3$) | 48.2% $\pm$ 2.9% |
| Mettenheim ($n = 1$) | 48.1% |
| Heppenheim ($n = 2$) | 46.4% $\pm$ 1.7% |
| Göcklingen ($n = 2$) | 45.6% $\pm$ 0% |

## 4. Discussion

### 4.1. Cross-Contamination

The study of fungi that may occur as secondary infections on grapes after a *Botrytis cinerea* infection is crucial for obtaining accurate biomass quantification results. There are many different fungal species potentially growing on *Vitis vinifera*, which could alter the quantification results [35]. This is particularly important when analysing different strains with varying growth characteristics. The Bc10nt primer used for qPCR was validated for cross-contamination with the three fungi *Penicillium expansum*, *Trichothecium roseum* and *Cladosporium* sp. After modifying the Bc10 primer set by adding an additional 10 bp following the genome sequence, no cross-contamination was detected. This demonstrates that the primer can successfully quantify *B. cinerea* biomass in field samples. However, it should be noted that grapevines can also be affected by other pathogens, such as *Aspergillus niger*. Analysing these organisms often requires a higher biosafety laboratory standard. Due to the extended primer pair, the likelihood of cross-contamination should be low. For example, studies in bunch trash showed no amplification of *Aspergillus* in qPCR [36] while using the primers of Suarez et al. [32].

### 4.2. Effect of Different Botrytis Strains on the Standard Calibration Curve

The qPCR analysis of various *B. cinerea* strains revealed only slight discrepancies between the number of spores counted and the number of spores quantified. For most strains, quantified biomass/spores were comparable to the manual cell count. The same strains were used as the qPCR standard curve when quantifying the biomass of the counted strains. The primer pair targets a highly conserved region in the genome [32]. However, there are large differences between some of the strains used. This may be due to limitations in cell

counting [37] or differences in the amplified region targeted by the primer. Comparing a cultivated *Botrytis* strain in the standard curve with unknown field samples is a typical approach, but could lead to inaccurate results.

### 4.3. qPCR as an Early Detection Method against Botrytis cinerea

When using qPCR, it is important to quantify the exact amount of biomass. Additionally, early detection of *B. cinerea* growth is crucial for monitoring pathogen growth in the vineyard. Measures can be taken to control the rot as soon as it is detected on the grapes. Currently, winegrowers rely on visual detection and often attempt to predict potential infections based on prior knowledge and previous occurrences. However, this method is subjective and unreliable, which can result in high crop losses. There are methods to control Botrytis using biocontrol agents [38], but the effectiveness is weather-dependent. The implementation of qPCR to detect *B. cinerea* before the onset of rot would enable better preparation and more effective monitoring of the grapes, while reducing treatment costs. The qPCR analysis revealed that *B. cinerea* can be detected up to 3–4 days before visible infection on the grapes. Subsequently, there was an increase on day 6, with no further increase on day 7, when sporulation became visible. The results add another layer to the potential of qPCR as an early detection method [36], providing more insight into the amount of time. It is important to note that this experiment was conducted in a laboratory setting, where growth conditions are optimal. Detection of infections in vineyards may be possible at an earlier stage if abiotic factors (e.g., temperature, humidity) slow down the growth of *B. cinerea*. Also, the stage of the berries has to be considered, since the inoculation, germination and growth of *B. cinerea* are influenced by ripening [38]. Pezet et al. [39] showed that the latent infection of *B. cinerea* is the same for resistant and susceptible grape varieties, but the growth inhibition is different. Therefore, the qPCR method presented could serve as a tool to assess the amount of time between detection and visual infection depending on the grape variety. This can be important for wine growers, who have to evaluate time and cost efforts during the harvest season.

### 4.4. PCR Method Validation—Size Ranges of Primer Sets and Composition

The results show promising ways to compare different strains of *B. cinerea* on a molecular biological level using simple repeat markers (SSRs). The SSRs amplified bands of the expected size (Table 5) with some variations from Fournier et al. [29]. Out of the 59 strains tested, both methods distinguished a significant proportion of the strains, with the capillary sequencing method distinguishing the most strains [35]. A typical agarose gel image can only differentiate the resulting bands of different strains up to 10 base pairs. Accurately distinguishing strains can be challenging due to issues with gel image resolution and band separation. Capillary sequencers can determine PCR products with up to 1 bp accuracy. The strains tested showed differences across all regions, indicating a high diversity of *B. cinerea* strains, adding to previous research [40,41].

The choice of method should be based on the importance of strain differentiation, the laboratory's budget and the operators' experience. For instance, a possible solution is to use an agarose gel for routine work or the pre-screening of strains, and then analyse the significant or indistinguishable strains with a capillary sequencer. The base pair range varied among the different primer pairs. Some primers exhibited low base pair variation, indicating a highly conserved target region. This demonstrates that not all primer pairs are equally important for strain differentiation.

### 4.5. Regional Differences between Strains

The methods tested can be used to gain further understanding of the diversity in every region as well as across different regions. Here, we showed that most of the regions tested had a variety of different strains. The only region where no differences were observed was Göcklingen, but the sample size was low ($n = 2$) and the samples were obtained in the same vineyard, making isolation of the same strain likely. Most of the strains were sampled in

the Rhine-Palatinate region (Edenkoben, Mußbach, Wollmesheim, Mörzheim, Göcklingen, Grünstadt). If regional populations exist, distant regions should show a greater diversity in comparison to the whole strain population (Table 7). While more distant regions like Bonn (>170 km), Freiburg (>150 km) and Ihrlingen (>150 km) showed greater diversity, this could not be observed in the Barsac (>800 km) and Mutzig (>100 km) region. Notably, the isolated strain in Bonn originated from a research area in the city, where no other vineyards were nearby. This might explain the different characteristics of the strain compared to a more vineyard-typical strain, since adaption and infection pathways are much harder. Interestingly, the DSMZ strain also showed a high difference from the strain population. Using the DSMZ strain as an indicator of the effectiveness of a treatment could be altered by regional strain behaviour. Using a "typical" strain for every region could lead to more accurate results. The results highlight the complexity of the pathogen.

## 5. Conclusions

The use of simple sequence repeat markers demonstrates that there are various methodological approaches for detecting and analysing different strains of *Botrytis cinerea*. All of these approaches are valid, depending on available equipment and trained personnel. Further research could utilise different strains to investigate their effects on grapes, must and wine. It is important to consider the impact of climate change on the aggressiveness of different *B. cinerea* strains. Therefore, further research is necessary to determine whether traditional approaches to *B. cinerea* are still applicable. Strain differentiation can aid in identifying differences between *B. cinerea* strains based on location, region, grape variety, or time, which can enhance our understanding of *B. cinerea* on grapevines. The locations of the genes targeted by the primers could provide insights into the attack patterns and survivability of different strains. This information could be used to develop strain specific treatments, focusing on more aggressive strains.

The qPCR method was validated for quantifying *B. cinerea* on grapes by eliminating secondary infection quantification. This opens up more research potential for *B. cinerea* field studies. qPCR is a highly sensitive method that can detect a *B. cinerea* infection at an early stage before it is visually detectable. This could prove useful for winemakers to obtain accurate and early results of an infection in the vineyard. It could be used to improve the assessment of fungicide application and fungal monitoring, for instance, the monitoring of vineyards for ice wine production, where grapes are even more susceptible to infection, and the loss of the entire harvest is possible. Here, time and the limits of detection are the most important factors. Further research should be conducted to gain more insight into the detection method for *B. cinerea* infection, such as using different aggressive strains to determine the detection limit.

**Supplementary Materials:** The following supporting information can be downloaded at: https://www.mdpi.com/article/10.3390/microbiolres15020037/s1, Figure S1: Map containing all locations where *Botrytis cinerea* strains were sampled from grape berries; Figure S2: Zoomed in map containing some of the locations, where *Botrytis cinerea* was sampled from grape berries. The map is zoomed in to give a more detailed view on strains from close regions; Figure S3: All strains were analyzed in order, followed by a negative control (example given); Figure S4: Example of the agarose gel. All strains tested are in order. Strain 1 followed by strain 2, 3,... in that order. At the end a negative control was done; Figure S5: Strain 1–10; Primer Bc7, Bc10; Figure S6: Strain 11–20; Primer Bc7, Bc10; Figure S7: Strain 21–30; Primer Bc7, Bc10; Figure S8: Strain 31–40; Primer Bc7, Bc10; Figure S9: Strain 41–50; Primer Bc7, Bc10; Figure S10: Strain 51–59; Primer Bc7, Bc10; Figure S11: Strain 1–10; Primer Bc1, Bc2, Bc4; Figure S12: Strain 11–20; Primer Bc1, Bc2, Bc4; Figure S13: Strain 21–30; Primer Bc1, Bc2, Bc4; Figure S14: Strain 31–40; Primer Bc1, Bc2, Bc4; Figure S15: Strain 41–50; Primer Bc1, Bc2, Bc4; Figure S16: Strain 51–59; Primer Bc1, Bc2, Bc4; Figure S17: Strain 1–10; Primer Bc3, Bc5, Bc6; Figure S18: Strain 11–20; Primer Bc3, Bc5, Bc6; Figure S19: Strain 21–30; Primer Bc3, Bc5, Bc6; Figure S20: Strain 31–40; Primer Bc3, Bc5, Bc6; Figure S21: Strain 41–50; Primer Bc3, Bc5, Bc6; Figure S22: Strain 51–59; Primer Bc3, Bc5, Bc6; Table S1: List of the locations, which are marked in the map of S1 and S2; Table S2: Overview of the band sizes analyzed with agarose gel electrophoresis and

ImageJ. Given are the analyzed strains and the primers used with their specific band sizes. Empty cells represent no detectable bands with the specific strain/primer.

**Author Contributions:** L.B. carried out the experiments. K.S. assisted L.B. with the qPCR experiments regarding cross-contamination and different strains for the standard calibration curve. L.B. wrote the manuscript with support from M.S.-S., A.J. and P.W.-H. conceived the original idea. M.S.-S. carried out the project administration and the funding acquisition. All authors have read and agreed to the published version of the manuscript.

**Funding:** This IGF Project (AiF 21630N) of the FEI is supported within the programme for promoting the Industrial Collective Research (IGF) of the Federal Ministry of Economic Affairs and Climate Action (BMWK), based on a resolution of the German Parliament.

**Data Availability Statement:** The original contributions presented in the study are included in the article/Supplementary Material, further inquiries can be directed to the corresponding author/s.

**Acknowledgments:** We thank Florian Schwander and Margrit Daum for help with the analysis on the capillary sequencer.

**Conflicts of Interest:** The authors declare no conflicts of interest. The funders had no role in the design of the study, in the collection, analysis or interpretation of the data, in the writing of the manuscript, or in the decision to publish the results.

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
