# Peer review of "Molecular Biological Methods to Assess Different Botrytis cinerea Strains on Grapes"

_2036-7481, doi:10.3390/microbiolres15020037_

Round 1
Reviewer 1 Report
Comments and Suggestions for Authors
The manuscript titled "Molecular biological methods to assess different Botrytis cinerea strains on grapes"
In this article the authors provide a good approach for the early detection of plant disease.
*The Introduction states the objectives of the work and provides an adequate background.
*The hypothesis of the study is clear.
*The Discussion explores the significance of the results of the work.
*Results are clear and concise.
However, the study requires some revision of the following issues.
1. At introduction section need to mention the need to early detection in paragraph.
2. In methods section please mention which parts of grapes used in the detection.
3. Conclusion need to be more summarized.
Author Response
Dear reviewer,
thanks for your comments.
Please see attached file for our answers.

Reviewer 2 Report
Comments and Suggestions for Authors
A well written and interesting manuscript. There are no flaws in experimental design and execution, and the discussion points are well-supported by the results. It is a practical study that is immediately useful to viticulture end-users.
line 146: insert space between "40" and "cycles"
In the methods section 2.2.8 specify what polymer was used in capillary electrophoresis, e.g. pop6 or pop7 etc. this is important as the different polymers affect the mobility of the fragments, and thus affects reproducibility.
line 325: italics on Aspergillus
line 398: likely mean "area" rather than "are"
In future work it would be interesting to include Botrytis pseudocinerea and the pathogenic “S strain” of B. cinerea
Author Response

(The authors gave the same response as above.)

Reviewer 3 Report
Comments and Suggestions for Authors
The subject of the study is original. The study is well written and discussed. However, Figure 6 in the discussion section should be placed in the results section. There are also a few spelling errors (Line 97,109 and 137 : Lipolytica ?, Line 271: agaJ ?). Statistical analysis methods ( (paired t-test and PCA ) should be added to the methods section
Author Response

(The authors gave the same response as above.)

Reviewer 4 Report
Comments and Suggestions for Authors
In this manuscript (microbiolres-2951432) entitled "Molecular biological methods to assess different Botrytis cinerea strains on grapes" submitted to Microbiology Research, Louis Backmann and colleagues have examined new methods, including strain differentiation and classification by simple sequence repeats (SSRs) and early detection of the fungus by qPCR, to detect Botrytis cinerea and differentiate strains. Collectively, these promising methods would facilitate the early detection of Botrytis cinerea strains. This research is interesting and convincing, but minor points need to be addressed to improve the quality of this manuscript.
1. For Table 1, authors should consider to show these sampling sites on a map in the revision.
2. For Figure 3, development of Botrytis cinerea on table grapes should be examined by microscope, and representative pictures should be included in the revised Figures.
3. For Figure 5, this picture is too dim to show details. Authors should consider to modify this picture in the revision.
4, Full names of abbreviations like PCR and qPCR in the abstract should be spelt out at their first appearance. Authors should check all abbreviations employed in the manuscript.
5. Authors need to standardize references according to the Microbiology Research template.
Author Response

(The authors gave the same response as above.)

Reviewer 5 Report
Comments and Suggestions for Authors
The authors present an interesting manuscript describing a new method of detection and differentiation of Botrytis cinerea strains. The early and accurate detection of this pathogen can help to mitigate the huge economic losses in grape production. The research is well planned with the use of appropriate methods. The introduction presents sufficient background information, methods are adequately described and results are well presented. The cited literature is properly selected for the content of the manuscript and the discussion properly explains the results in the view of published data. I do however miss the supplementary material which should include fully described results from capillary gel electrophoresis and agarose gel electrophoresis.
Concluding I recommend this manuscript to be accepted for publication in Microbiology research after minor revisions.
In-text comments:
Line 2: Please do not separate words in the title.
Line 100: Please add the table caption and the accuracy of geographical coordinates
Line 101: Format Table 1 according to the formatting of the remaining tables.
Line 108: Please use dot “.” As decimal separator and use coma as thousand separator
Line 110: the species name should be written in lowercase
Line 117: Please provide a citation for this information if this data was previously published, or established for this manuscript. If you have compared different isolation protocols and have them compared please include this data if available as a supplement.
Line 227: Please increase the with of the x and y axes as well as the font size.
Line 131:various -> seven
Line 155: Often ethanol sterilization is insufficient for plant surface sterilization, consider additionally using an oxidizing sterilizer. Did you confirm that sterilization was successful and did you wash out the remaining ethanol with sterile water?
Line 256: Did you confirm the assumptions for pairwise t-tests?
Line 257: p should be in italics
Line 259: Remove the framing in this figure
Line 280: Why there is no information on the band size for Bc9 primers on an agarose gel?
Line 281: Please include this data for all primers in the supplementary materials.
Line 285: Please include these graphics in supplementary data with marked columns and band sizes.
Author Response

(The authors gave the same response as above.)

Reviewer 6 Report
Comments and Suggestions for Authors
This manuscript is devoted to the fungus Botrytis cinerea, which causes diseases of various plant species worldwide. It causes, among other things, large economic losses in vineyards. The manuscript mainly contains important methodological aspects using molecular biology. The authors characterized these methods well and presented subsequent stages of the procedure using various strains of B. cinerea from many regions of Europe and selected other fungi that can secondarily infect vine berries. The authors use Simple Sequence Repeat Markers for detecting and analyzing different strains of Botrytis cinerea. It has been shown that qPCR is a highly sensitive method that can detect a B. cinerea infection at an early stage before symptoms can be visually detected. It was indicated how the results of the current work can be used to the advantage of vineyard owners. This work should be published in Microbiology Reserarch/MDPI, but before that, numerous, mainly mycological, errors should be corrected in the manuscript, which are indicated in Remarks.
Remarks
Line 29 'Botrytis spp. is a well-known pathogen', if you write spp. (= species plurum) it means that you mean numerous species from the genus Botrytis, so Plural should be used
Line 30 'Botrytis cinerea spp.' - this entry is incorrect; spp. must be removed
Line 31 the abbreviation B. allii, B. byssoides… should be used throughout the manuscript.
Line 33 such as gladioli – not clear
Line 91 'Mußbach, Bonn, Mußbach' – Mußbach is given twice
Line 95 – please provide an explanation or further DSMZ data
Line 97 Lipolytica -should be lowercase
Line 98 spp. – it should be non-italic
Line 100 Table 1 – write the title of the table, what it contains
Line 100 Table 1 Location (add N, E ?)
Line 102 2.1.2. – complete the purposes for which it is done
Line 110 Lipolytica –it should be lower case
Line 124 Botrytis it should be in italic
Line 126 spp. it should be not italic
Line 137 Lipolytica – must be in lower case
Line 146 40cycles – add space
Line 217 it should be Cladosporium spp. - spp. not italic and one dot at the end of the sentence
Line 229 spp. – it should be not italic. This should also be changed inside Figure 1 (twice) and throughout the manuscript. Moreover, please take into account: if you mean one unknown species of Cladosporium - then it should be written Cadosporium sp., but if you think that these were different species of the genus Cladosporium, then you should write Cladosporium spp. In Line 318 you write “the three fungi ……..” – this means that you mean one species of Cladosporium sp.
Line 240 cell number – why do you write a spore number in one place and cell number in others – should there be a spore number everywhere?
Line 255 please note that the infection process by B. cinerea takes little time, after which the incubation process begins
Line 260 it should be in this case Botrytis cinerea (Bc) as explanation to Figure
Line 286 Figure 6 – is it possible to lighten it?
Line 325 Aspergillus – it should be italic
Line 352 consider revising this sentence
Line 435 consider revising this sentence
Line 457 consider revising this sentence
References – bibliography is provided in an inconsistent manner, incompatible with MDPI guidelines
Comments on the Quality of English Languagesee Remarks
Author Response

(The authors gave the same response as above.)
